# ReNeg and Backseat Driver: Learning from demonstration with continuous human feedback

## Abstract

In autonomous vehicle (AV) control, requiring an agent to make mistakes, or even allowing mistakes, can be quite dangerous and costly in the real world. For this reason we investigate methods of training an AV without allowing the agent to explore and instead having a human explorer collect the data. Supervised learning has been explored for AV control, but it encounters the issue of the covariate shift. That is, training data collected from an optimal demonstration consists only of the states induced by the optimal control policy, but at runtime, the trained agent may encounter a vastly different state distribution and has seen little relevant training data. To mitigate this issue, we intentionally have our human explorer make sub-optimal decisions. In order to have our agent not replicate these suboptimal decisions, supervised learning requires that we either erase these actions, or replace these action with the correct action. Erasing these actions is wasteful and replacing these actions is difficult, since it is not easy to know the correct action without driving the car. Since supervised learning falls short, we introduce an alternate framework that includes continuous scalar feedback on each action, marking which actions we should replicate, which we should avoid, and how sure of this we are. Our framework, called "ReNeg", learns from from human demonstration and human evaluative feedback, collected entirely before any training begins. Our agent learns continuous control from sub-optimal behavior, but sub-optimal behavior that is safely performed by a human. We find that a human demonstrator can explore sub-optimal states in a safe and controlled manner, while still getting enough gradation in the states to benefit learning. The way we collect data and the algorithm we use to compute feedback on the data we call "Backseat Driver." Backseat Driver gives us state-action pairs matched with scalar values representing the score for those action in those states. We call the more general learning framework ReNeg, since it learns a regression from states to actions given negative as well as positive examples. We empirically validate several models in the ReNeg framework, testing on lane-following with limited data. We find that the best solution is a generalization of mean-squared error and outperforms supervised learning on the positive examples alone.

## 1 Problem Formulation

We seek a way learn autonomous vehicle control using only RGB input from a front-facing camera. (Example input image in the appendix.) Specifically, we want to produce a policy network to map states to actions in order to follow a lane as close to center as possible. We would like the agent to behave optimally in dangerous situations, and recover to safe situations. For this to happen, the data the agent trains on must include these sub-optimal states. However, we do not want to leave the safety determination up to any machine. Thus, we require that a human explorer take all the actions. This is referred to as "Learning from Demonstration" or LfD. An expert human does the exploration and thus can limit the danger by controlling how dangerous the negative examples are (i.e. the human can swerve to show suboptimal driving but never drive off the road, as an RL agent likely would). Still, in order to get into sub-optimal states, the explorer will need to take sub-optimal actions. Ideally we could replace these sub-optimal actions with labels corresponding to the correct

and optimal actions to take so we could perform supervised learning, which provides the strongest signal. However, it is notoriously difficult to assign supervised labels to driving data because it is hard to know the correct steering angle to control a vehicle when your steering has no effect on what happens to the vehicle (Ross et al., 2012). Instead of trying to enable humans to assign supervised labels by somehow showing the consequences of their actions, we focus on letting humans assign feedback that evaluates the sub-optimal actions, without actually representing the optimal action.

Thus our goal is to learn a deterministic continuous control policy from demonstration, including both good and bad actions and continuous scores representing this valence. Our problem statement is somewhere in between supervised learning and RL: we focus on a general approach that is capable of mapping continuous sensor input to an arbitrary (differentiable) continuous policy output and capable of using feedback for singular predetermined actions. Our problem falls within the supervised setting since our agent cannot chose want actions to take, in order to avoid the agent exploring dangerous states, and all data collection is done prior to training. However, we also wish to incorporate evaluative scalar feedback given to each data point, which traditionally falls under the RL framework. We refer to this problem setting as the ReNeg framework, since we are essentially performing a regression with scalar weights attached to each data point, that can be positive and negative. For the demonstration, a human demonstrator drives, yielding state-action pairs the agent can learn from. In order to teach the agent which actions are good or bad, an expert critic, or "backseat driver," labels the actions with a continuous scalar.

For a viable solution to this problem, we need to define a loss function that induces an effective and robust policy. We also need to choose what driving data to collect to expose the agent to a variety of good and bad states, and in particular show the agent how to get out of bad states where an accident is imminent. What data to collect is non-obvious: we want to explore a range of good and bad states so the agent learns a reasonable policy for how to act in all kinds of states. However, to make this feasible in the real world, we want to avoid exploring dangerous states. Finally, we need to carefully choose a way to collect human feedback that contains signal from which the agent can learn, but is not too hard to collect. We will discuss the choices we made for these three parts of the problem in the Loss Function, Driving Data, and Feedback sections respectively. Although we validate in simulation, specifically Unity, for the sake of ease, the algorithms we test could be easily trained in the real world.

## 2 RELATED WORK

Learning from demonstration has mostly been studied in the supervised learning framework and the Markov Decision Process (MDP) framework. In the former, it is generally known as imitation learning (or behavioral cloning in the AV setting), and in the latter, it is generally known as apprenticeship learning. There are other relevant RL algorithms, but none for LfD in our problem setting, as we will show.

The supervised learning in this area has focused on a least squares regression that maps from an input state to action. The easiest approach to take here is to have an expert perform an optimal demonstration, and then simply use that as training data. The main issue here is that the runtime and training distributions can be vastly different. That is, once the trained agent is acting on its own after training and encounters states it has not seen, it does not know how to act, and strays further from the intended behavior. This problem is known as the Covariate Shift and the widely accepted solution generally follow the approach laid out in DAgger Ross et al. (2010). DAgger allows the agent to explore and then uses the expert to label the new dataset, then training on all of the combined data. Such an approach has even been improved upon both to address scalar feeddback by incorporating experts that can label $Q$ values with the AggreVaTeD algorithm (Ross & Bagnell, 2014), and to and to address deep neural networks by finding a policy gradient with the Deeply AggreVaTeD algorithm (Sun et al., 2017). However, these DAgger based policies require the agent to ex-plore and make mistakes.

The MDP research on apprenticeship learning has largely been on inverse reinforcement learning, or IRL (Abbeel & Ng, 2004), in which a reward function is estimated given the demonstration of an expert. However, often in this framework, the reward function is restricted to the class of linear combinations of the discrete features of the state, and the framework only allows for positive examples. In addition, there has been work on inverse reinforcement learning from failure (Shiarlis

et al., 2016), which allows for a sequence of positive or a sequence of negative examples. There is also distance minimization for reward learning from scored trajectories (Burchfiel et al., 2016), which allows for gradation in the scores, but does not allow for an arbitrary reward function on continuous inputs or labels for atomic actions as opposed to a trajectory or sequence of actions. Moreover, these IRL methods are not a candidate for our problem, since they require an exact MDP solution with a tractable transition function and exploration to find the optimal policy. The issue we have is not that we don't have the reward function, but that even with the more informative feedback, we cannot use exploration to learn the optimal policy.

It is interesting to note that there are off-policy RL algorithms as well, but we would like to highlight that this is not the same thing as LfD. LfD, as we use it, means that we have collected all of our data before training. This could be thought of as on-policy only if our policy and start state never changes. Whereas, off-policy RL (e.g. Off-Policy Actor-Critic(Degris et al., 2012), Q-Learning(Watkins & Dayan, 1992), Retrace(Munos et al., 2016)) generally requires agents to have a non-zero probability of choosing any action in any state, in order for the algorithm to converge. (Moreover, it is the somewhat parenthetical opinion of at least one of authors of this paper that in the off-policy policy gradient RL framework, using importance sampling to calculate an expectation over a different action distribution is fine, but changing the objective function from an expectation over the learned policy state-visitation distribution to an expectation over the behavior (exploratory) state-visitation distribution, as in (Degris et al., 2012), is an unsatisfactory answer that does not give the optimal policy for the agent when running on the learned policy, as we will want to do.) Normalized Actor Critic (NAC) does attempt to bridge the gap between off-policy and LfD, and works with bad as well as good demonstration, however, NAC does not allow for restricted exploration either, since it adds entropy to the objective function to encourage exploration (Gao et al., 2018). NAC has also only been done for discrete action control, not continuous as we want to do.

Finally, there are many RL algorithms that use human evaluative feedback, but none for LfD. One RL algorithm with human feedback of note is COACH (MacGlashan et al., 2017), which is based on the stochastic policy gradient. COACH is an on-policy RL algorithm that uses human-feedback to label the agent's actions while exploring. COACH's view on human feedback helps us to draw connections to RL, as we will discuss later. However, COACH was designed for on-policy exploration and uses discrete feedback values of 1 and -1, whereas we generalize to continuous values in $[-1, 1]$. We cannot use a stochastic policy gradient in a justified manner, since we do not explore with a stochastic policy.

Out of all the LfD work in the AV context, the most notable has either been on behavioral cloning (Bojarski et al., 2016) (Pan et al., 2017) or using IRL to solve sub-tasks such as driving preferences that act on top of a safely functioning trajectory planner (Kuderer et al., 2015). To the best of our knowledge, no research so far has focused on using any kind of evaluative scalar feedback provided by a human in the context of AV control with LfD. That is, no one has solved how to take states, actions, and continuous feedback with respect to those actions, and convert them into a control policy for an AV, without having the AV explore. We believe that this is a major oversight: many AV research groups are investing huge amounts of time into collecting driving data; if they used our model, they could improve performance simply by having an expert labeler sit in the car with the driver for no additional real time.

## 3 OUR APPROACH

### 3.1 LOSS FUNCTION

Let $\theta$ be the steering angle in the demonstrated example; $\hat{\theta}$ the steering angle predicted by the policy network (or PNet); $D$ the absolute difference between $\theta$ and $\hat{\theta}$, $|\theta - \hat{\theta}|$; and $f_\theta$, or $f$, the feedback for the demonstrated angle. We collect $f \in [-1, 1]$. The loss function we choose should have the following 3 properties:

1. Minimizing the loss should minimize the D for positive examples. That is, when $f > 0$, $\frac{\partial Loss}{\partial D} \geq 0$.

2. Minimizing the loss should maximize the distance between $\theta$ and $\hat{\theta}$ for negative examples. That is, when $f < 0$, $\frac{\partial Loss}{\partial D} \leq 0$.

3. The rate at which the loss is minimized should be determined by the magnitude of the feedback. That is, when $f > 0$, $\frac{\partial |Loss|}{\partial f} > 0$, and when $f < 0$, $\frac{\partial |Loss|}{\partial f} < 0$.

These three properties together ensure that the network avoids the worst negative examples as much as possible, while seeking the best examples. Given an input state $s$, the first loss function that comes to mind is what we term "scalar loss":

$$Loss_{scalar} = f_s * (\theta_s - \hat{\theta}(s))^2$$

This loss function is notable for several reasons. First, it is a generalization of mean squared error, the standard behavioral cloning loss function:

$$Loss_{MSE} = (\theta_s - \hat{\theta}(s))^2$$

Mean squared error is a well-principled loss function if you assume Gaussian noise in your training data. That is, you assume the probability of your data can be given by Gaussian noise around some mean and you learn to predict $\hat{\theta}$ as that mean. Given this assumption, you can derive MSE as the loss that produces a maximum likelihood estimate for your parameters. Let the parameters of the model be represented by $p$ and probability be $Pr$. $\hat{\theta}$ is parameterized by $p$ and will be used interchangeably with $\hat{\theta}_p$, when clarity is needed. Please note that $\theta$ refers to the angle label, and not the model parameters:

$$\text{argmax}_p Pr(data)$$

$$\text{argmax}_p \prod_{\theta \in data} Pr(\theta|\hat{\theta}_p)$$

$$\text{argmax}_p \sum_{\theta \in data} log(Pr(\theta|\hat{\theta}))$$

$$\text{argmax}_p \sum_{\theta \in data} log(Pr(\theta|\hat{\theta}))$$

$$\text{argmin}_p \sum_{\theta \in data} -log(Pr(\theta|\hat{\theta}))$$

$$Loss = -log(Pr(\theta|\hat{\theta}))$$

$$Loss = -log(\frac{e^{-\frac{(\theta-\hat{\theta})^2}{2\sigma^2}}}{\sqrt{2\pi\sigma^2}})$$

$$Loss = \frac{(\theta - \hat{\theta})^2}{2\sigma^2} - log(\sqrt{2\pi\sigma^2})$$

Generally, $log(\sqrt{2\pi\sigma^2})$ is left out since it is a constant w.r.t. our parameters and so will go away when the gradient is taken, leaving:

$$Loss = \frac{(\theta - \hat{\theta})^2}{2\sigma^2}$$

Generally, $2\sigma^2$ is also left out, since it only acts to scale the gradient, and can be accounted for by adjusting the learning rate of gradient descent, leaving:

$$Loss = (\theta - \hat{\theta}_p)^2$$

However, we note that, if we interpret $|f|$, the magnitude of our feedback, as $\frac{1}{2\sigma^2}$, we can view $|f|$ as a measure of certainty. This certainly applies to a Gaussian distribution with a variance of at least $\frac{1}{2}$, since $f \in (0, 1]$ for positive examples. For negative examples, we generalize further by removing the magnitude calculation, and allowing our feedback to be negative. This enforces that we minimize the probability of negative data:

$$Loss = f * (\theta - \hat{\theta}_p)^2$$

To be able to easily recover behavioral cloning, we introduce two hyperparameters. The first such parameter is the ability to threshold feedback values. If we threshold, we simply replace every

$f$ with $sign(f)$. Thresholding eliminates gradations in positive and negative data. Additionally, we introduced the parameter $\alpha$, which scales down all our negative examples' feedback: $f := max(f, \alpha f)$. This trades off between behavioral cloning and avoiding negative examples. We apply $max(f, \alpha f)$ after we threshold, so if we threshold with and set $\alpha$ to 0.0, we recover behavioral cloning.

Our scalar loss is also notable since it closely resembles a loss that induces a stochastic policy gradient. In a standard RL policy network such as REINFORCE, the gradient would be $\nabla_p = \nabla_p(Q^\pi(\theta) * -log(Pr(\theta)))$ (Williams, 1992). The loss then, that would induce this gradient is $Loss = Q^\pi(\theta) * -log(Pr(\theta))$. $R$, the return, or a sample contributing to $Q^\pi(\theta)$, is generally used. In continuous control, one could instead predict a mean $\hat{\theta}$ for a normal distribution and then sample your action $\theta$ from that normal distribution. As demonstrated above, if you replace $log(Pr(\theta))$ with the probability density function for a normal distribution, the loss you wind up with is precisely MSE scaled by $R$. Substituting this scalar into the derivation above at every step, you get:

$$Loss = R * (\theta - \hat{\theta})^2$$

A full derivation of this loss given the stochastic policy gradient and Gaussian policy can be found in the appendix. Clearly if we view $f_\theta$, our feedback, as $R_{\hat{\theta}}$ and assume a Gaussian policy, we get our scalar loss function. COACH in fact points out that you can view online feedback given for the current policy, $f_{\hat{\theta}}$ as a sample from $Q^\pi(\hat{\theta})$, or $A^\pi(\hat{\theta})$, the advantage function, and empirically verifies that this works for on-line RL with discrete feedback (MacGlashan et al., 2017). This similarity was useful for inspiration and ideation, but actually falls short of rigorous justification for two reasons. Since we are training off-policy, and, more specifically, on a pre-determined policy that does not explore stochastically and does not vary depending on the current predicted policy, we run into major issues justifying our loss this way.

First, the main "off-policy" issue here is that for the stochastic policy gradient to hold, the exploration must be stochastic. However, in our case, the data is drawn from a pre-determined, deterministic policy. We can illuminate the intuition for why the stochastic policy gradient no longer works by considering a simple example. Consider the network attempting to learn the correct predicted $\hat{\theta}$ for a given state. Consider that there is only one demonstrated $\theta$, -1, with a feedback of -1. Now, no matter what $\hat{\theta}$ the network predicts, the $\theta$ action that is taken during training will always be -1, and the feedback will always be -1. Moreover, consider that the actual optimum is $\hat{\theta} = 1$, and the network is currently predicting $\hat{\theta} = -2$. Using our scalar loss, the network will increase distance between $\theta$ and $\hat{\theta}$ by decreasing $\hat{\theta}$, making the policy worse and worse. This would not happen when using a stochastic RL policy, since the network can explore states around the current predicted $\hat{\theta}$ by choosing appropriate actions.

Using a stochastic RL policy, given enough samples on either side of $\hat{\theta}$, the network will have larger and larger gradients, the more negative $\hat{\theta}$ is. But these gradients will not keeping "pushing" the prediction to the left of -1, but rather will randomly cause $\hat{\theta}$ to move around, gradually moving to the right as it finds better feedback, and eventually converging at the optimum of $\hat{\theta} = 1$. The network will move less "violently" and more "stably" the closer $\hat{\theta}$ gets to 0. And when the network eventually reaches the positive numbers, it might get "pulled" to the left a bit when it happens to sample a worse action, but it will not get pulled as strongly as when it samples a better action to the right. We can now intuitively see the issue: the neural network cannot not influence the probability of seeing an example again, which can lead to problems with learning the policy. In RL, a policy network can try a bad action, and then move the policy away from that action and not revisit it. On the other hand, if we have a bad example in our training set for a given state, on every epoch of training, our neural net will encounter this example and take a step away from it, thus pushing our network as far away from it as possible. Taking these steps is not necessarily helpful since the network may not have favored taking the bad action before.

We could use some sort of importance sampling (Silver et al., 2014) (Degris et al., 2012), as is done with stochastic off-policy exploration, to scale down the loss for examples we are far away from. However, this would make our update have almost no effect when we are far away from positive examples, and with the deterministic exploration of ReNeg, the distinction between positive and negative examples now matters. We can't have it both ways just by multiplying by the probability of $\hat{\theta}$ given our model. (This happens since the Gaussian PDF decreases exponentially with difference

$|\theta - \hat{\theta}|$, but the loss only increases quadratically with the difference, due to the logarithm. Thus the gradient tends toward 0 due to the differing rates of growth, as the difference gets large.) Moreover, importance sampling consistently reduced performance for learning from demonstration in the NAC paper (Gao et al., 2018).

Another, perhaps less significant issue, is that our feedback represents $Q*$, and not $Q\pi$. Even if the human critic could re-assigned labels as the agent trains, he/she would have no way to sample the return from $Q\pi$, without letting the agent explore freely. This is significant because an action that is optimal for the expert may be a dangerous and poor decision for a non-optimal policy. This is perhaps less significant than the other issue, since there are no actions in our data that could be considered both good and risky. What is good for one policy (i.e. steering back to the middle of the road) is generally good for all policies, and so can act rather greedily with respect to $Q*$, even though it will not always be following the optimal policy.

For these reasons, we find the comparison to stochastic policy gradients useful, but ultimately un-compelling. Applying RL losses to supervised learning does not provide the mathematical justification we need. The stochastic policy gradient is no longer computing the gradient of the current policy at all. Thus, we focus primarily on extending and generalizing MSE. Yet, we can still learn from the policy gradient comparison. In particular, we acknowledge that the sign of our feedback is far more significant than it was in the RL context (which is one of the reasons we introduce the $\alpha$ parameter, which scales down the importance of negative examples, in the event that we collect too many negative examples). In RL, using the stochastic policy gradient, it did not matter if negative examples "pulled" $\hat{\theta}$ closer to them, so long as the positive examples pulled $\hat{\theta}$ more, since you would eventually try one of those actions, if it is a nearby optimum. However, since actions are no longer sampled dependent on the current policy, suddenly the sign matters very much. In fact, even if we have a positive example for a state, if we have more negative examples than positive examples, we may wind up ignoring our positive examples entirely in an effort to get away from our negative examples. This case highlights the trouble inherent in using negative examples: It is hard to know how and when to take into account the negative examples.

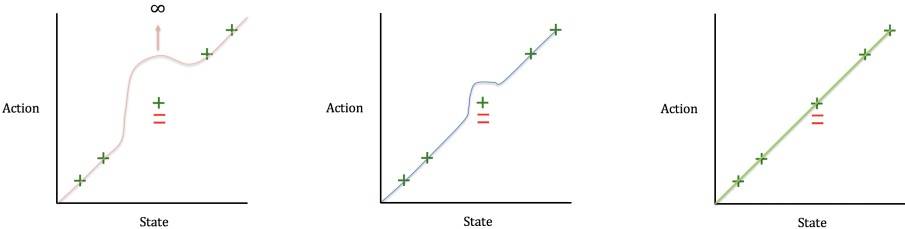

Figure 1: Potential outcomes of regression with negative examples

Now, let us consider negative examples in the ReNeg framework. For example, as shown in Figure 1, if we perform a regression on positive and negative examples with more negative examples than positive in one state, we may wind up in a case where our loss is minimized by a prediction of positive or negative infinity, and thus our regression is "overwhelmed" by the negative examples. This led us to our second "exponential" loss:

$$Loss_{exp} = |\theta(s) - \hat{\theta}(s)|^{2f}$$

Using this loss, negative examples will have infinite loss at distance 0, and then drop off expo-nentially with distance. We hope that this will create regressions more akin to the second image in Figure 1. In this image, adding more negative points will still nudge the regression away more and more, but one positive point not too close by should be enough to prevent it from diverging to positive or negative infinity. It should be noted that the loss in a particular state still could only have negative examples, especially in a continuous state-space like ours where states are unlikely to be revisited. However, the reduction in loss caused by diverging to infinity would be so small that it should not happen simply due to continuity with nearby states enforced by the structure of the network. In addition, one concern with this loss could be that for positive fractional differences, and negative non-fractional differences, the desired property 3) of loss functions no longer holds. That

is, our positive loss will not grow with $f$ if the difference being exponentiated is a fraction. And for negative exponents, the loss will only grow if the difference is a fraction that shrinks as it is raised to increasing powers of $f$. However, we hope that for negative examples, distances that are more than 1 unit away will not occur often (since 1 unit is half the distance range). We discuss a potential future solution in the appendix to patch this loss function.

Our final loss function should produce regressions more like the final image in Figure 1: We propose directly modelling the feedback with another neural network (which we call the FNet) for use as a loss function for our PNet. If this FNet is correctly able to learn to copy how we label data with feedback, it could be used as a loss function for regression with our PNet. Thus, in order to maximize feedback, our loss function would be as follows:

$$Loss_{FNet} = -FNet(s, \hat{\theta})$$

After learning this FNet, we can either use it as a loss function to train a policy network or, every time we want to run inference, we can run a computationally expensive gradient descent optimization to pick the best action. Because the latter does not depend on the training distribution (so we do not have the issue of the runtime and training distributions being different), and it is more efficient, we choose an even easier version of the latter: we pick the best action out of a list of discrete options according to the FNet's predictions. One feature of the FNet is that adding more negative points will not "push" our regression further away from this point, but rather just make our FNet more confident of the negative feedback there. This may not be the desired effect for all applications. Moreover, the FNet cannot operate on purely positive points with no gradation. That is, behavioral cloning cannot be recovered from it.

### 3.2 DRIVING DATA

We recorded 20 minutes of optimal driving and labeled all of this data with a feedback of 1.0. Choosing the suboptimal data and how to label it was a bit more tricky. One reason that we are not using reinforcement learning is that letting the car explore actions is dangerous. In this vein, we wanted to collect data only on "safe" driving. However, the neural network needs data that will teach it about bad actions well as good actions that recover the car from bad states. In order to explore these types of states and actions, we collected two types of bad driving: "swerving" and "lane changing". The first image in Figure 2 is swerving. In swerving, the car was driving in a "sine wave" pattern on either side of the road. We collected 10 minutes of this data on the right side of the road and 10 minutes on the left. The second image is lane changing. For this, we drove to the right side of the road, straightened out, stayed there, and then returned to the middle. We repeated this for 10 minutes, and then collected 10 minutes on the left-hand side as well.

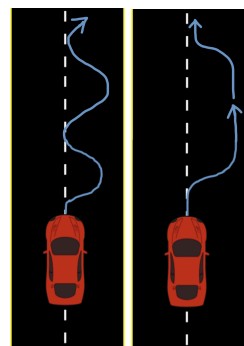

Figure 2: Types of driving: swerving (left), lane change (right)

### 3.3 "BACKSEAT DRIVER" FEEDBACK

Backseat Driver is the framework we use to compute and collect feedback in the AV context. Our feedback includes much more information than just a reward (as is used in RL): we take our label to directly measure how "good" an action is relative to other possible actions. We use this approach instead of labeling rewards for actions both because we found it an easy way to label data with feedback, and because it contains more signal. How exactly to label the data with feedback, however, is non-obvious. At first, we considered labeling using a slider from -1 to 1. However, using a slider can be non-intuitive in many cases and there would be discontinuities in feedback you would want to give. For example, if the demonstrator is driving straight off the road and then starts to turn left back onto the road, there would be a large discontinuity in the very negative and then slightly positive feedback.

In order to circumvent these issues, and to make the labeling process more intuitive, we decided to collect feedback using the steering wheel. We found that it is easier for people to focus on the steering angle, since that is how we are used to controlling cars. Our first thought was to just turn the steering wheel to the correct angle. However, this is very difficult to estimate, especially on turns,

when you cannot see the actual effects your steering is having on the car. (Note that if we did this, our algorithm would turn into behavioral cloning.) Instead, we decided to label the differential. That is, we turned the wheel to the left if the car should steer more left. This signal shows where the error is (i.e. "You should be turning more" or "You're turning the wrong way"). Note that the label does not need to be the exact correct angle; it just needs to show in which direction the current action is erring, and proportionally how much. We call this method of human feedback collection "Backseat Driver." In order to process the angle labels into a feedback value in $[-1, 1]$, we used the equation below:

FEEDBACK$(c, \theta)$

1   **if** $\text{sign}(c) == \text{sign}(\theta)$ or $|c| \leq \epsilon$
2       **return** $1 - |c|$
3   **else**
4       **return** $-|c|$

Note: We first normalize all of our corrections by dividing by the greatest collected correction $c$, so all of our $c$ values fall in $[-1, 1]$. In line 1 above, if we are turning the steering wheel in the same direction as the car (with some $\epsilon$ of error), then the feedback should be positive. (We set epsilon to $\frac{5}{\theta_{max}}$ so that it allows up to 5 degrees of tolerance.) Since $c$ represents a delta in steering, a greater delta should result in a less positive signal. Therefore, the feedback should be proportional to $-|c|$. We add 1 to ensure all $c$ in the same direction as $\theta$ are positive. If $c$ is in a different direction than we were steering (line 3), then the feedback should be negative, so we just return $-|c|$ as the feedback. Thus the greater the delta, the more negative the feedback will be. If $c$ is in the same direction, on the other hand, we chose to scale these feedbacks up so that the feedback is positive, but less positive for a greater differential. This makes sense since if, for example, the car is steering left and we tell it to steer more left, this is not as bad as if the car is steering the wrong way. Thus, slow actions back to the center of the road will be rewarded less than quick actions back to the center of the road.

### 3.4 ARCHITECTURE

We chose to use only an hour of data because we wanted to see how far we could get with limited data. While our feedback is relatively easy to collect compared to other options, it still takes up human hours, so we would like to limit its necessity. We sampled states at a rate of approximately two frames per second, since states that are close in time tend to look very similar. We augmented our data by flipping it left-right, inverting the angle label, and leaving the feedback the same. After this augmentation, we had 17,918 training images and 3,162 validation images (a 85:15 split).

We chose to learn our policy with an end-to-end optimization of a neural network to approximate a continuous control function. Such networks are capable of tackling a wide array of general problems and end-to-end learning has the potential to better optimize all aspects of the pipeline for the task at hand. Given the option to use separate (differentiable) modules for sub-tasks such as computer vision, or to connect these modules in a way that is differentiable for the specific task, the latter will always perform better, since the end-to-end model always has the ability to simply not update the sub-module if it will not help training loss.

We used transfer learning to help bootstrap learning with limited data for both the PNet and Fnet. We decided to start with a pretrained Inception v3 since it has relatively few parameters, but has been trained to have strong performance on ImageNet, giving us a head start on any related computer vision task. We kept some early layers of Inception and added several of our own, followed by a tanh activation for the output. We tried splitting the network at various layers and found that one about halfway through the network (called mixed_2) worked best. The layers we added after the split were fully connected layers of sizes 100, 300, and 20. For the FNet, the angle input is concatenated onto the first fully-connected layer. (Find an architecture diagram in the appendix.)

## 4 EXPERIMENTS

We tested our trained models by running them in our Unity simulator and recording the time until the car crashed or all four tires left the road. (Note: We designed the ReNeg framework so it need

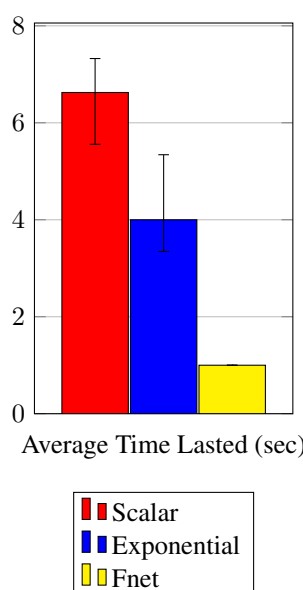

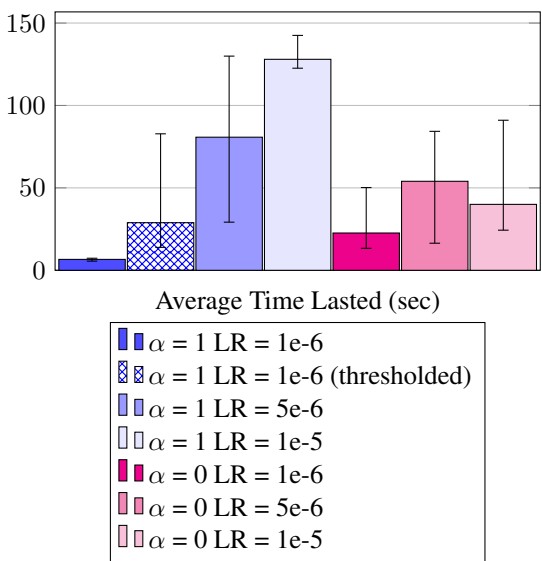

Figure 3: Initial comparison of loss functions.

Figure 4: Scalar loss function performance including negative examples vs. only positive examples

not be run in a simulator. This would work just as well in the real world. We only used a simulator because it was all we had access to.) We first tested our PNet with the scalar loss, our PNet with the exponential loss, and our FNet. We plotted their mean times over the eight runs with standard deviation. See Figure 3 below. (All of these were on the default hyperparameters listed in the appendix, except for the exponential loss model, for which we set $\alpha$ to 0.1 since otherwise we could not get it to converge.) Based on the predicted angles, the FNet seemed to primarily predict feedback based on state, not angle; this makes sense given that the feedback in "bad" states is generally "bad", except for the split second when the "good" action takes place.

Given that the scalar loss performed best (and was training correctly), we spent more time tuning the hyperparameters for this model. This can be seen in Figure 4. We note several interesting things from figure 4 exploring the scalar loss. First, the pink group is identical to the solid blue group except that all the $\alpha$ values are 0, meaning that negative-feedback examples are zeroed out and effectively ignored in training. The blue bars are much higher than their pink counterparts, indicating that the negative data is useful. Second, we can see that thresholding the feedback to -1 and 1 (the blue crosshatch pattern) increased the scalar performance to about 30 seconds (compared to 6 seconds with the default hyperparameters). At first this could be taken to indicate that having gradations in positive and negative data could be harmful to training. However, we see that the same performance is achieved (and surpassed) by increasing the learning rate instead of thresholding, by looking to the 2 rightmost blue bars. The reason for this is likely that we tuned the learning rate to work well on thresholded data, and so, when we don't threshold/clone our data, the scalars on the loss drop significantly, forcing the network to take smaller steps, and effectively decreasing the learning rate. Increasing the learning rate instead of thresholding yields much better performance, indicating that gradations in the data (with a high learning rate) do help training.

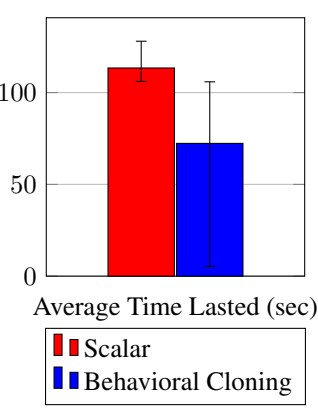

Figure 5: Over 3 training runs, our scalar loss model performed over 1.5 times as well as the behavioral cloning benchmark, with significantly less variance.

Note, $\alpha$ can in fact be in $[0, \infty]$, however we focused on $\alpha = 0$ and $\alpha = 1$, since this corresponds to no negative examples and an equal weighting between positive and negative examples. It it difficult to tune independently due its relationship with the learning rate.

Moreover, all we are trying to show is that using negative examples, with some relative importance to the positive examples, can be beneficial.

After selecting the best learning rate for each model, we then trained new versions of each network 2 more times, for a total of 3 models each, to account for stochasticity in SGD. (The best learning rate for both turned out to be 1e-5; see the appendix for more details on tuning.) Each time, we let the model drive the car for 8 trials and calculated the performance as the mean time before crashing over these 8 trials. We then calculated the mean performance for each over the 3 training sessions. Figure 5 shows the results.

## 5   CONCLUSION

We hypothesized that for the task of learning lane following for autonomous vehicles from demonstration, adding in negative examples would improve model performance. Our scalar loss model performed over 1.5 times as well as the behavioral cloning baseline, showing our hypothesis to be true. The specific method of regression with negative examples we used allows for learning deterministic continuous control problems from demonstration from any range of good and bad behavior. Moreover, the loss function that empirically worked the best in this domain does not require an additional neural network to model it, and it induces a stochastic policy gradient that could be used for fine-tuning with RL. We also introduced a novel way of collecting continuous human feedback for autonomous vehicles intuitively and efficiently, called Backseat Driver. We thus believe our work could be extremely useful in the autonomous control industry: with no additional real world time, we can increase performance over supervised learning by simply having a backseat driver.

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

# 6 APPENDIX

## 6.1 EXAMPLE INPUT

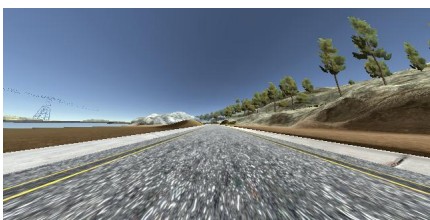

## 6.2 ARCHITECTURE

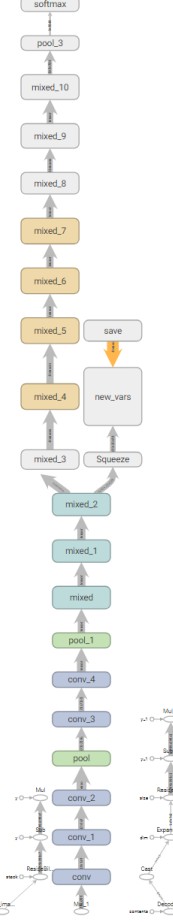

The architecture of our model, based on Inception v3. After the split, the branch on the right is the branch used for our PNet or FNet, while the left branch is ignored. The new layers we added were 3 fully connected layers of sizes 100, 300, and 20. The final activation is Tanh to ensure the output range of (-1,1). For the FNet, the angle input is concatenated onto the first fully-connected layer.

## 6.3 HYPERPARAMETERS

Our batch size was 100 and we trained for 5 epochs. Unless otherwise specified for a given model in the experiments, we used an $\alpha$ value of 1.0, we did not threshold, and we used a learning rate of 1e-6. As in the Inception model we were using, our input was bilinearly sampled to match the

resolution 299x299. Likewise, we subtracted off an assumed mean of 256.0/2.0 and divided by an assumed standard deviation of 256.0/2.0.

## 6.4 TRAINING METRICS

During training, we kept track of two validation metrics: the loss for the model being trained, and the average absolute error on just the positive data multiplied by 50. The first we refer to as "loss" and the second we refer to as "cloning error" (since it is the 50 times the square root of the cloning error) or just "error". The reason we multiplied by 50 is that this is how Unity converts the -1 to 1 number to a steering angle, so the error is the average angle our model is off by on the positive data. (This is true with the maximum angle set to 50.)

During training, these two metrics generally behaved very similarly, however, in the models for which we increased the learning rate, these eventually start to diverge. In this case, the error on the positive data started to increase, but the loss was still decreasing. For this reason, we tried varying the learning rate on several models, to see if the loss was more important than the "cloning" error. It is clear that the behavioral cloning models (thresholded with $\alpha = 0.0$) should in general do better on the "cloning" error, since they are very closely related. Whereas for non-thresholded data, it was trained with examples weighted differently. And for the negative data, it was trained to also get "away" from negative examples. We hope that even though the cloning error may increase, this means that it is because the model is choosing something better than (yet further away from) the positive examples. We still use the cloning error, however, because it is a useful intuitive metric for training and comparison.

## 6.5 LEARNING RATE TUNING

We tried several learning rates for both behavioral cloning and ReNeg. We compared the performance, shown in figures 6 and 7, and found that 1e-5 worked best for both.

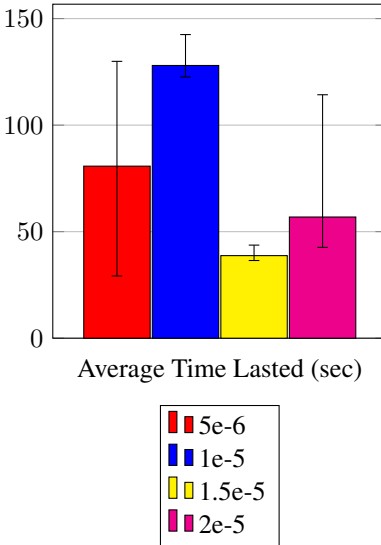

Figure 6: The scalar loss performed best with a learning rate of 1e-5.

## 6.6 GAUSSIAN POLICY GRADIENT DERIVATION

Here is the derivation from the stochastic policy gradient to the loss that induces it, which is very similar to our scalar loss:

$$\nabla Loss = \nabla(R * -log(Pr(\theta)))$$

$$\nabla Loss = \nabla(R * -log(\frac{e^{-\frac{(\theta-\hat{\theta})^2}{2\sigma^2}}}{\sqrt{2\pi\sigma^2}}))$$

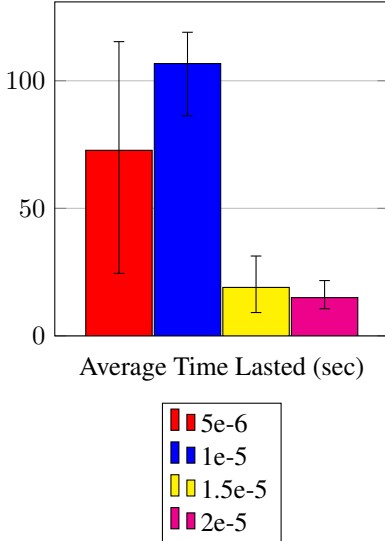

Figure 7: The behavioral cloning loss performed best with a learning rate of 1e-5.

$$\nabla Loss = \nabla(R * -(log(e^{-\frac{(\theta-\hat{\theta})^2}{2\sigma^2}}) - log(\sqrt{2\pi\sigma^2})))$$

$$\nabla Loss = \nabla(R * -(log(e^{-\frac{(\theta-\hat{\theta})^2}{2\sigma^2}})))$$

$$\nabla Loss = \nabla(R * \frac{(\theta-\hat{\theta})^2}{2\sigma^2})$$

$$\nabla Loss \propto \nabla(R * (\theta-\hat{\theta})^2)$$

$$Loss \propto R * (\theta-\hat{\theta})^2$$

### 6.7 FUTURE RESEARCH

Future research involving human feedback should focus on 2 things: the loss function and enforcing continuity. These will be briefly explored in the next two sections.

Note, before we discuss potential extensions of this work, that fine-tuning the policy once it is acceptably safe is a separate but also interesting problem. Supervised approaches involving a safety driver taking over control, and retraining on this data a la DAgger, should probably be explored. Additionally, instead of aggregating the new data with the old, active learning approaches could be explored, where the model is not entirely retrained. we point the reader to (Pan et al., 2017) for an AV application of DAgger.

#### 6.7.1 LOSS FUNCTION

Immediate next steps should likely focus on alterations to the loss function. Here we introduce a fourth desired property that can ensure our negative examples have less or the same "impact" as they get farther away, and positive examples have more of an impact as they get farther away. In other words, for positive examples, as D increases, the update (derivative) is always the same or greater in magnitude, and for negative examples, the same is true as D decreases.

4. Concavity:

The loss is concave up with respect to D i.e. $\frac{\partial^2 Loss}{\partial D^2} >= 0$.

For positive examples, when $f > 0$: This enforces that $|\frac{\partial Loss}{\partial D}|$ has have a minimum at $D = 0$ (the optimum), since it must be positive and increasing. Ideally, $\frac{\partial Loss}{\partial D}$ would equal 0 at $D = 0$, and nowhere else.

For negative examples, when $f < 0$: This enforces that $|\frac{\partial Loss}{\partial D}|$ has have a minimum a $D = \infty$ (the optimum), since it must be negative and increasing. Ideally, $\frac{\partial Loss}{\partial D}$ would approach 0 as $D \to \infty$, and be 0 nowhere else.

We propose two loss functions that meet properties 1-4:

1. We can accomplish this exponential decay by modifying our scalar function in a very easy way: Move the "sign" of f into the exponent:

$$Loss_{inverse} = |f| * (\theta(s) - \hat{\theta}(s))^{2*sign(f)}$$

Using this loss function we have all three properties satisfied. That is, positive examples encourage moving towards them, negative examples encourage moving away from them, and the amount of this movement increases with the magnitude of f. Moreover, we also have the property that, in negative examples, loss drops off exponentially with the distance from the negative example (because we are dividing by it).

2. If we want our scalar loss function to have neither an exponential decay nor an exponential increase with the distance from the negative points, we can simply use the following loss:

$$Loss_{absolute} = f * |\theta(s) - \hat{\theta}(s)|$$

This has the not-so-nice property that, in the positive example, it allows outliers much more easily than the traditional squared loss. However, it has the very nice property that, given a single state input, as long as you have more positive examples than negative examples, your loss will always be minimized in that state by a value between your positive examples. This is because, as soon as you get to your greatest or least positive example, every step away from your positive examples will cost you 1 loss, for each positive example you have, and you will only lose 1 loss for each negative example you have. (Note, if you are not thresholding, then this translates to more total $|f|$ for positive examples than negative examples.)

### 6.7.2 Continuity

Because in both our scalar and exponential loss, our loss function at a given state with just a negative example is minimized by moving away from the negative example, our regression in that state will tend toward positive or negative infinity. Certainly having a cost on negative examples that drops of exponentially will help, but it may not be enough. Moreover, we may not want to rely on the structure of neural networks to discourage this discontinuity. Therefore, research could be done on adding a regularization term to the loss that penalizes discontinuity. That is, we would add some small loss based on how dissimilar the answers for nearby states are. Of course, this implies a distance metric over states, but using consecutive frames may suffice.

