# OpenReview forum: "ReNeg and Backseat Driver: Learning from demonstration with continuous human feedback"
_ICLR.cc/2019/Conference_

### Official Review · AnonReviewer1 · 2018-11-02
**Very confusingly written with unclear experiments.**

**Rating:** 2
**Confidence:** 5

**Review:**

Summary:

This paper proposes a method to get feedback from humans in an autonomous vehicle (AV). Labels are collected such that the human actually moves a steering wheel and depending on the steering wheel angle disagreement with the direction the vehicle is actually moving a feedback value is collected which is used to weight the scalar loss function used to learn from these demonstrations.

Experiments on a simple driving simulator is presented.

Comments:

I think this paper is attempting to address an important problem in imitation learning that is encountered quite often in DAgger, AggreVate and variants where the expert feedback is provided on the state distribution induced by the learnt policy via a mixture policy. In DAgger (where the corrections are one-step as opposed to AggreVate where the expert takes over and shows the full demonstration to get Q(s,a)) it is difficult to actually provide good feedback especially when the expert demonstrations are not getting executed on the vehicle and hence hard for humans to ascertain what would be the actual effect of the actions they are recommending. In fact there is always a tendency to overcorrect which leads to instability in DAgger iterations.

The paper proposes using a modified feedback algorithm on page 6 whose magnitude and sign is based on how much the correction signal is in agreement or disagreement with the current policy being executed on the vehicle.

Unfortunately this paper is very confusingly written at the moment. I had to take multiple passes and still can't figure out many claims and discussions:

- "To the best of our knowledge, no research so far has focused on using any kind of human feedback in the context of AV control with learning from demonstration" - This is not true. See:

"Learning Monocular Reactive UAV Control in Cluttered Natural Environments, Stephane Ross, Narek Melik-Barkhudarov, Kumar Shaurya Shankar, Andreas Wendel, Debadeepta Dey, J. Andrew Bagnell, Martial Hebert" who used DAgger for autonomous driving of a drone with human pilot feedback.

- Lots of terms are introduced without definition or forward references. Example: \theta and \hat{\theta} are provided early-on are refered to on page 3 in the middle of the page but only defined at the end of the page in 3.1.

- Lots of confusing statements have been made without clear discussion like "...we could also view our problem as a contextual bandit, since the feedback for every action falls in the same range..." This was a baffling statement since contextual bandit is a one-step RL problem where there is no credit assignment problem unlike sequential decision-making settings as being dealt with in this paper. Perhaps something deeper was meant but it was not clear at all from text.

- The paper is strewn with typos, is really verbose and seems to be written in a rush. For example, "Since we are off-policy the neural network cannot not influence the probability of seeing an example again, and this leads can lead to problems."

- The experiments are very simple and it is not clear whether the images in figure 2 are the actual camera images used (which would be weird since they are from an overhead view which is not what human safety drivers would actually see) or hand-drawn illustrations.

---

> ### Author Response · Authors · 2018-11-18
> **Clarification**
>
> Hi Reviewer 1,
> Thanks for your feedback. We would like to clarify and rebut some of your points:
>
> "To the best of our knowledge, no research so far has focused on using any kind of human feedback in the context of AV control with learning from demonstration" - This is not true. See: "Learning Monocular Reactive UAV Control in Cluttered Natural Environments, Stephane Ross, Narek Melik-Barkhudarov, Kumar Shaurya Shankar, Andreas Wendel, Debadeepta Dey, J. Andrew Bagnell, Martial Hebert" who used DAgger for autonomous driving of a drone with human pilot feedback.
> Their definition of “feedback” is very different from ours. They used “feedback” to mean some sort of indication to the human demonstrator about how the huma’s actions would turn out. This is an interesting problem, to solve, since it is hard, for example, to control a car without visual and consequential feedback, but it is entirely different than the evaluative feedback that our human critic provides to the agent. Perhaps we need to further clarify what we mean by feedback.
> They were not only doing “learning from demonstration”, as we framed the problem. They allowed the agent to freely explore and then took over in emergencies. This is too risky for autonomous vehicles. Instead, we only allowed our human explorer to safely take sub-optimal actions. And this only happens before any training begins. Perhaps we need to further clarify what we mean by learning from demonstration.
>
> Our simulator is a perspective simulation, not top down. We will include an image in the final version.
> Our agent is entirely off-policy. But not only that, no exploration is done at all past the initial collection. Really we are generalizing MSE, not COACH or any stochastic policy gradient, since our approach is not calculating a stochastic policy gradient at all. Our approach is really just MSE generalized with a scalar feedback. We did, however, realize one very important difference from stochastic policy gradients by considering them: the sign of our feedback is very important now and can very easily ruin convergence if we collect more negative examples in a state than positive examples. This is not a feature of the stochastic policy setting. For this reason, we introduce the alpha parameter, which suddenly becomes very relevant in the LfD setting, which again, has no exploration.
> We also provide a we provide an algorithm for converting angle to feedback and empirically validate it.

---

> > ### Comment · AnonReviewer1 · 2018-11-25
> > **Important paper**
> >
> > Thanks for all the clarifications. I do believe that this is an important paper tackling an important problem in imitation learning state-of-the-art in robotics domains. The comments are very clear and insightful while the paper needs quite a bit of work in presentation and experiments. I encourage the authors to incorporate the comments in the paper to make it clearer.

---

### Official Review · AnonReviewer2 · 2018-11-02
**Interesting topic, but poorly communicated and lacking novelty**

**Rating:** 4
**Confidence:** 4

**Review:**

Despite many high profile successes in research, DeepRL is still not widely used in applications. One reason for this is that current methods typically assume the agent can learn by exploring many states and actions, however, in many real world tasks, such as driving used here, poor actions can be dangerous. Therefore, methods that can provides the flexible benefits of RL while avoiding this are of significant interest, one promising general ideas pursued for this has been to use human demonstrations.

A number of approaches to Inverse RL have been studied, but many make the assumption that the demonstrations are all examples of optimal behavior. This can be challenging if, for example, some examples contain suboptimal behavior, and it also means that the agent does not get to observe non-optimal trajectories and how to correct for them; the resulting policy often performs poorly due to the distributional shift between the demonstration trajectories and the trajectories induced by the learned policy.

This work attempts to correct for these problems by labeling the demonstration actions between $[-1, 1]$ indicating how good or bad the demonstration actions are. This introduces a challenge for learning, since good actions can be copied, but a bad action is more ambiguous: it does not necessarily imply the action are far away from the bad action is a good action.

One view of this work is that they introducing 3 losses for behavior cloning with weighted labels: A weighted (based on the label) L2 loss, an exponential loss and directly fitting the loss and searching over a discrete set of actions to find the highest estimate weighting. Note the current equation for $Loss_{FNET}$ doesn't make sense because it simply minimizing the output of the network, from the text it should be something like $(f - \hat{\theta})^2$?

The text discusses why rescaling the negative examples may be beneficial, but as far as I can tell, figure 4 you only consider $\alpha=\{0, 1\}$? Based on the text, why weren't intermediate values of $\alpha$ considered?

It could benefit from copy-editing, checking the equations and in some cases describing concepts more concisely using clear mathematical notation instead of wordy descriptions that are difficult to follow.

``Thus the assumption
that our training data is independent and identically distributed (i.i.d.) from the agent’s encountered
distribution goes out the window'' This is a misleading statement regarding the challenge of distributional shift in off-policy RL. The challenge is that state distribution between the behavior policy and the learned policy may be quite different, not just not iid. Even in on-policy RL the state visitation is certainly not usually iid.

``In the off-policy policy gradient RL framework, this issue is typically circumvented by changing the
objective function from an expectation of the learned policy’s value function over the learned policy
state-visitation distribution to an expectation of the learned policy’s value function over the behavior
(exploratory) state-visitation distribution (Degris et al., 2012). In the RL framework, this could be
dealt with by an approximation off-policy stochastic policy gradient that scales the stochastic policy
gradient by an importance sampling ratio (Silver et al., 2014) (Degris et al., 2012). ''. The importance sampling in Degris is not to correct for the objective being under the behavior state policy and DPG (Silver et al., 2014) specifically does not require importance sampling so it shouldn't be referenced here. This paragraph seems to be conflating two issues: the distributional shift between the behavior state distribution and the policy state distribution that can make off-policy learning unstable, and importance sampling to estimate outcome likelihoods using behavior experience.

This work is on a very important topic. However, in its current form it is not well-communicated. Additionally, the best performing method is not novel (as the author's state $\alpha=1$, the best performing setting, is essentially the same as COACH but with scalar labels). For these reasons reason, I think this work may be of limited interested.

---

> ### Author Response · Authors · 2018-11-18
> **Clarification**
>
> Hi, thank you for your feedback. I want to provide some clarification on a few of your points.
>
> Re your confusion about our FNet loss: we used the output of the FNet as an adversarial loss for our PNet. This didn't end up working as well as other losses.
>
> The fact that examples are not i.i.d. and this *is* a known issue for supervised learning. From the DAgger paper itself, “A typical approach to imitation learning is to train a classifier or regressor to predict an expert’s behavior given training data of the encountered observations (input) and actions (output) performed by the expert. However since the learner’s prediction affects future input observations/states during execution of the learned policy, this violate the crucial i.i.d. assumption made by most statistical learning approaches.”
>
> And finally, you're correct that our algorithm was essentially the same as COACH with scalar feedback values except that 1) we don’t use eligibility traces, 2) we show it works off-policy, 3) we provide a method for collecting feedback in AV context, and 4) we tested other algorithms, this one just happened to work the best.

---

> > ### Author Response · Authors · 2018-11-18
> > **Follow-up**
> >
> > Hi Reviewer 2,
> > Two more points to add:
> > The citation is not wrong. The DDPG does an amazing job of summarizing off policy approaches.
> > We did consider intermediate values of alpha. Just didn’t include the large graph in final edit. We will put it back.

---

> > > ### Comment · AnonReviewer2 · 2018-11-26
> > > **Improved, but still hard to read and of limited interest**
> > >
> > > I have read the updated version, it is improved and addresses some of the issues I raised.
> > >
> > > However, it still its challenging to read and still probably of interest to only a small fraction of the community.
> > >
> > > It still uses long paragraph to explain things that would be better explained by writing specifically the equations/losses in use and makes sub-optimal use of space.
> > >
> > > For example, there half a page in the main text to deriving the L2 loss from assumption of Gaussian noise (a very well-known result), but the specific loss used to train FNet is not given (I'm assuming L2 loss).
> > >
> > > In the revised version you specifically state you believe the off-policy RL objective is problematic (this is correct, it is problematic for the reasons you describe), but then there is no comparison empirically to an off-policy RL method. It is challenging for any method, given only the labelled demonstrations, to generalize to novel states ... you don't demonstrate that this approach here does better.
> > >
> > > I remained puzzled that the paper talks about the importance of scaling the loss of the negatively labelled examples but the only values of alpha in experiments are 0 and 1 (effectively no scaling, or ignoring negative labels altogether).
> > >
> > > In many cases Q^\pi is typeset as Q\pi.

---

> > > > ### Author Response · Authors · 2018-11-26
> > > > **Response - Could be More Polished but Important Work**
> > > >
> > > > Hi,
> > > >
> > > > To answer some of your points:
> > > >
> > > > 1. There are a couple typos we missed and perhaps could have written things more concisely, however we feel that this does not entirely negate the ideas and work we are contributing. Sure, the paper format could be more polished, but we believe this type of learning - from previously collected data with scalar feedback on the actions - represents an important framework in between supervised learning and RL that is not well studied. Moreover, it is very applicable to both AV and could be a very beneficial approach in this field. Additionally, although we do not explore other applications of the framework, we believe it could be applicable to other areas as well. For example, learning how much medication to administer from previously collected treatments by doctors with scalar evaluative outcomes. In the AV field, companies that want to train their AV safely can have a human do the demonstration and simply add another human to evaluate the actions, and get a drastic benefit from this over the standard end-to-end approach of behavioral cloning. We admit the paper could be more polished,  but feel that it still constitutes work of interested to the AV and ML field despite the typos.
> > > >
> > > > 2. Yes, Mean Squared Error was used to train the FNet
> > > >
> > > > 3. The typical off-policy approaches from RL do not work for us because they generally still require exploration. It was our explicit goal to avoid exploration. For example, Importance sampling for the stochastic policy gradient approach still requires stochastic exploration. Since we are modeling something akin to the Q* directly with f, our FNet is analogous to a deterministic policy gradient, which can be used for off-policy RL. However, again, we want to stress that Off-policy RL requires a non-zero chance of choosing any given (state, action) pair to prove convergence. Since we are clearly not doing this, we can find no real justification by drawing the RL connection. The only reason we mention it is 1) For inspiration and insight into what doesn’t work. 2) Because our policy is justified in terms of a Gaussian model, a Gaussian stochastic policy gradient would wind up looking very similar and likely could be used for fine tuning with little need to tweak hyper-parameters. However, since we do not demonstrate the latter, we don’t belabor this point.
> > > >
> > > > 4. We now mention in the paper why we only use alpha of 0.0 and 1.0. Namely that all we were trying to show is that some alpha worked and every adjustment of alpha requires re-tuning the learning rate. Moreover the alpha and threshold generalization provided an easy way to recover. behavioral cloning. Perhaps we should have also stressed the fact that our training data already contained more positive examples than negative, since we included data in which we followed the lane optimally. Thus, we already had more positive examples and the training turned out fine. Clearly, however, it would be very important if one had collected twice the amount of negative data as positive data, for the reasons discussed in the loss function section. Then our alpha parameter allows for this to be easily corrected.

---

> > > > > ### Comment · AnonReviewer2 · 2018-11-26
> > > > > **Short response**
> > > > >
> > > > > Just to respond to the last 2 points quickly.
> > > > >
> > > > > 3. You have expert data along with scalar labels of the actions. This could be used for off-policy RL directly (e.g. Q-Learning). In this case, one is hoping the function approximator may learn to generalize. Here, with FNet is used to learn to label actions, and the hope it is generalizes. In both cases, you are trying to make use of function approximator generalisation.
> > > > >
> > > > > 4. Your text talks about scaling the negative labels, but the two choices of alpha correspond to either ignoring negative examples or scaling them equally. This introduces a mismatch between the textual claims and the actual experiments.

---

> > > > > > ### Author Response · Authors · 2018-11-27
> > > > > > **Reply**
> > > > > >
> > > > > > 3) If we view our feedback f as Q* values (as we intended), then sure, our FNet already is a Q-Net. If we view our feedback f as reward, r, then try to do Q learning, we need to calculate the TD error. That is, the target value would need to be r+max_a'[Q(s',a')]. This max becomes very tricky in continuous space. (The one model I know about that does this, NAF, places restrictions on the shape of the Advantage function in order to calculate this max.)  For this reason, and motivated by how COACH interprets human feedback, we thought this was unnecessary.
> > > > > >
> > > > > > 4) Alpha is a parameter in our model that we used to scale by 0 or 1. We didn't have time to manually tune the LR for other values of alpha. Other values may be important if you have more negative data than positive, but we did not. We added in a paragraph briefly mentioning this, but we could have been more clear. Still, we maintain that adding in the negative examples, at some scale, was useful. (Despite the fact that we did not need to decrease their importance.)

---

### Official Review · AnonReviewer3 · 2018-11-02
**This paper addresses the problem of applying reinforcement learning in cases where exploration is too dangerous. The authors presented an algorithm that collects driver data and solicits human feedback during operations, hence the name "Backseat driver." They demonstrate benefit in collecting both negative examples (examples of bad driving) and positive examples.**

**Rating:** 3
**Confidence:** 4

**Review:**

This paper is clearly written and identifies an important point that exploration is dangerous in the autonomous driving domain. My key objection to this paper is that, even though the method is intended to deal with problems where exploration is dangerous and therefore should not be done, the method relies on negative examples, which are presumably dangerous. If simulations are used to generate negative examples and those are used, then the benefit of the presented method over standard reinforcement learning goes away.

I have several questions/comments/suggestions about the paper:

1. Can one perhaps present only mildly bad examples (e.g., mild swerving) to reinforcement learning in a way where the algorithm can understand that significant swerving, like what is shown in figure 2, is even worse?
2. The backseat driver feedback described seems to granular. I think that, to be realistic, the algorithm should allow for feedback that is less precise (e.g., turn further, turn the other way), without requiring information on proportions.
3. Please add an architecture diagram.
4. In figure 4, what is the difference between the first and fourth items? They have exactly the same description in the legend.
5. The experiments are not convincing. They lead one to conclude that negative examples are beneficial, which is good, but not surprising. Because negative examples are generated, a comparison with regular reinforcement learning should be done.

---

> ### Author Response · Authors · 2018-11-09
> **Clarification**
>
> Hi, thank you for your feedback. I want to provide some clarification, since I see there were some misunderstandings about our paper.
>
> Re your key objection to the paper: the point is that we have a human explore instead of the agent, so we can control what kind of states are explored. We explore states that are bad but not dangerous, so the agent learns and can extrapolate what kinds of states are bad. You cannot do this with normal agent exploration as in RL, because the agent might explore dangerous states. And you cannot do this with imitation learning, because that framework doesn't allow for any suboptimal states to be encountered.
>
> And re your point about simulation: we did use a simulator to test this algorithm, but we didn't NEED to. This algorithm could be done in real life; we just chose to test in a simulator because that was all we had on hand.
>
> 1. Yes, this is the point of our paper. All the examples we show the agent are only mildly bad.
> 2. That could be interesting to try, too, but using the granular feedback worked for us.
> 4. Typo, thanks for catching that. Item #4 should show LR = 1e-5.
> 5. RL is not a good comparison for our algorithm, because the problem we're trying to solve is learning from driving in the real world. RL involves dangerous exploration. Imitation learning (which we compared against) is the standard way to learn from real-world demonstration.

---

> > ### Author Response · Authors · 2018-11-18
> > **Follow-up**
> >
> > Hi Reviewer 3,
> > Just a few more clarifications:
> > Our advantage over RL is that an expert human does the exploration and so can limit the danger. Our expert driver can control how dangerous the negative examples are. We never drove off the road, as an RL agent would. The whole point was to not explore with RL.

---

### Public Comment · ~Raviteja_Chunduru1 · 2018-11-03
**Enquiring about dataset**

Interesting paper. I would like to explore this paper more. Can you please send me the dataset (swerving and lane change data) to my email, or direct me to where I can find it? My email: ravi.tej.310@gmail.com

---

### Meta-Review · Area_Chair1 · 2018-12-12

**Confidence:** 5
**Recommendation:** Reject

**Metareview:**

The authors consider the interesting and important problem of how to train a robust driving policy without allowing unsafe exploration, an important challenge for real-world training scenarios. They suggest that both good and intentionally bad human demonstrations could be used, with the intuition being that humans can readily produce unsafe exploration such as swerving which can then be learnt using both positive and negative regressions. The reviewers all agree that the paper would not appeal to or have relevance for the wider community. The reviewers also agree that the main ideas are not well presented, that some of the claims are confusing, and that the writing is not technical enough. They also question the thoroughness of the empirical validation.